# *Bifidobacterium animalis* ssp. *lactis* MG741 Reduces Body Weight and Ameliorates Nonalcoholic Fatty Liver Disease via Improving the Gut Permeability and Amelioration of Inflammatory Cytokines

**DOI:** 10.3390/nu14091965

**Published:** 2022-05-07

**Authors:** Moon Ho Do, Mi-Jin Oh, Hye-Bin Lee, Chang-Ho Kang, Guijae Yoo, Ho-Young Park

**Affiliations:** 1Research Division of Food Functionality, Korea Food Research Institute, Wanju 55365, Korea; do.moon-ho@kfri.re.kr (M.H.D.); mjoh@kfri.re.kr (M.-J.O.); 50023@kfri.re.kr (H.-B.L.); gjyoo@kfri.re.kr (G.Y.); 2Central Research Institute, MEDIOGEN Co., Ltd., Jecheon 27159, Korea; changho-kang@naver.com

**Keywords:** *Bifidobacterium animalis* ssp. *lactis* MG741, nonalcoholic fatty liver disease, fasting hyperinsulinemia, gut health, gut–liver axis

## Abstract

Diet-induced obesity is one of the major causes of the development of metabolic disorders such as insulin resistance and nonalcoholic fatty liver disease (NAFLD). Recently, specific probiotic strains have been found to improve the symptoms of NAFLD. We examined the effects of *Bifidobacterium animalis* ssp. *lactis* MG741 (MG741) on NAFLD and weight gain, using a mouse model of high-fat-diet (HFD)-induced obesity. HFD-fed mice were supplemented daily with MG741. After 12 weeks, MG741-administered mice exhibited reduced fat deposition, and serum metabolic alterations, including fasting hyperinsulinemia, were modulated. In addition, MG741 regulated Acetyl-CoA carboxylase (ACC), fatty acid synthase (FAS), sterol regulatory element-binding protein 1 (SREBP-1), and carbohydrate-responsive element-binding protein (ChREBP) expression and lipid accumulation in the liver, thereby reducing the hepatic steatosis score. To determine whether the effects of MG741 were related to improvements in gut health, MG741 improved the HFD-induced deterioration in gut permeability by reducing toxic substances and inflammatory cytokine expression, and upregulating tight junctions. These results collectively demonstrate that the oral administration of MG741 could prevent NAFLD and obesity, thereby improving metabolic health.

## 1. Introduction

The increasing number of obese people worldwide is a major public health problem. Obesity causes diverse metabolic disorders such as impaired glucose and lipid homeostasis, inflammation, hypertension, and hepatic steatosis [1,2,3,4]. The incidence of nonalcoholic fatty liver disease (NAFLD) in obese people is greater than 75% [5]. Obesity causes hepatic insulin resistance by increasing mitochondrial reactive oxygen species production and the release of free fatty acids, which leads to the accumulation of fatty acids in the liver, thereby inducing NAFLD [6,7,8].

Several researchers have demonstrated that the gut–liver axis plays an important role in the development of NAFLD [9]. High-fat-diet (HFD)-induced changes in gut microbial composition increase pathogen-associated molecular patterns, such as lipopolysaccharides (LPS), and gut permeability, thereby resulting in metabolic endotoxemia [10]. Since the liver is provided more than 50% of its blood by the splanchnic district, it is one of the organs most exposed to gut-derived toxins [11]. Therefore, increased endotoxin in circulating blood causes insulin resistance, inflammation, and fat accumulation, and can eventually progress to NAFLD [12]. It has also been shown that HFD-fed germ-free mice are resistant to developing obesity, insulin resistance, and NAFLD [13]. Therefore, improvements in gut health play an important role in suppressing these metabolic diseases.

Bifidobacteria are considered one of the important health-beneficial bacteria in the intestinal tract, and are present at high abundance within the newborn gut during breastfeeding [14]. Many preclinical and clinical studies have shown the probiotic activities and beneficial effects of bifidobacteria, such as improvements in gut health and the prevention and treatment of diverse metabolic disorders [15,16]. Our previous study reported that *Bifidobacterium animalis* ssp. *lactis* MG741 (MG741), isolated from human infant feces, exhibited strong antioxidant activities such as nitric oxide, 2,2-diphenyl-1-picrylhydrazyl (DPPH) free radical scavenging and 2,2′-azino-bis-3-ethylbenzothiazoline-6-sulfonic acid (ABTS) radical scavenging [17]. As a further study, here, we aimed to determine whether MG741 can improve metabolic disorders in HFD-fed mice.

## 2. Materials and Methods

### 2.1. Materials

Fluorescein isothiocyanate-dextran (FITC-dextran; MW 4000) was purchased from Sigma-Aldrich (St. Louis, MO, USA). Primary antibodies against Acetyl-CoA carboxylase (ACC), fatty acid synthase (FAS) and sterol regulatory element-binding protein 1 (SREBP-1) were obtained from Abcam (Cambridge, MA, USA), and carbohydrate-responsive element-binding protein (ChREBP) and glyceraldehyde 3–phosphate dehydrogenase (GAPDH) were obtained from Cell Signaling Technology (Beverly, MA, USA). Secondary antibody against rabbit was purchased from Thermo Fisher Scientific (Waltham, MA, USA).

### 2.2. MG741 Preparation and Culture Condition

MG741 was isolated from healthy breast-fed infant feces and identified [17]. MG741 was cultured and maintained in De Man, Rogosa and Sharpe (MRS) broth at 37 °C. Cultures were started from frozen stocks stored at −80 °C in MRS broth (BD, Franklin Lakes, NJ, USA) containing 20% glycerol. The powder of MG741 was provided from MEDIOGEN Co., Ltd. (Jecheon, Korea).

### 2.3. Animals and Dosage Information

Male 8-week-old C57BL/6J mice were obtained from Central Lab Animal Inc. (Seoul, Korea). Mice were randomly allocated into four groups (*n* = 9) and acclimatized in a Specific Pathogen Free (SPF) animal facility with a 12 h/12 h light/dark cycle and an average room temperature of 22 ± 2 °C. All mice received diet ad libitum for food and energy intake. The experimental protocol was approved by the Animal Welfare Committee of the Korea Food Research Institute (KFRI-M-20013). The normal diet (ND) group was fed a Teklad global 2018s diet and the HFD group was fed a 60% kcal from fat diet (Teklad TD.06414). MG741-treated groups were fed the same diet as the HFD group but concomitantly were orally gavaged with a daily dose of MG741 (1.0 × 10^5^ or 1.0 × 10^6^ CFU/day/mouse).

### 2.4. InAlyzer Analysis

At week 12, the body composition of the mice was analyzed with an InAlyzer (Medikors Inc., Seongnam, Korea). The anesthetized mice were placed in the InAlyzer and then whole body scanning was performed to measure body and fat mass. The body composition was divided into three types, lean tissue (blue color), medium-density fat (yellow color), and high-density fat (red color). The scanning images and parameters were acquired by InAlyzer software.

### 2.5. Biochemical Analysis

At the end of the experiment, 12 h-fasted mice were anesthetized and euthanized. Then, tissues were collected and immediately stored at −80 °C until use. Blood was collected from the abdominal vein and serum was separated using centrifugation. The obtained serum was stored at −80 °C until further processing. Serum alanine aminotransferase (ALT) and aspartate aminotransferase (AST) concentrations were estimated using kits from Cusabio (CSB-E16539m, CSB-E12649m, Wuhan, China). Liver triglyceride and serum total cholesterol, high-density lipoprotein (HDL) cholesterol, and low-density lipoprotein (LDL) cholesterol were measured with kits from Abcam (ab65336 and ab65390). Serum leptin concentration was determined using kits from R&D systems (MOB00B, Minneapolis, MN, USA), and insulin and endotoxin levels were determined with a kit from Thermo Fisher Scientific (EMINS, A39553). All experiments were performed as per the manufacturers’ instructions and the homeostasis model assessment of insulin resistance (HOMA-IR) index was calculated as HOMA-IR = glucose (mmol/L) × insulin (mU/L)/22.5 [18].

### 2.6. Western Blotting

Total proteins were obtained using protein lysis buffer (PRO-PREP^TM^, iNtRON Biotechnology, Seongnam, Korea) containing phosphatase inhibitors. The lysates were separated by SDS-PAGE and transferred onto PVDF membranes. After blocking using 5% skimmed milk, membranes were incubated with primary antibodies (against SREBP1, ChREBP, ACC, FAS, and GAPDH) at 4 °C overnight and then with secondary antibody for 1 h at 25 °C. The EZ-Western Lumi Femto Kit (DoGenBio Co. Ltd., Seoul, Korea) was used to detect the protein bands, and their intensities were quantified by a ChemiDoc XRS+ imaging system (Bio-Rad, Hercules, CA, USA).

### 2.7. Quantitative Real-Time PCR

Total RNA of colon tissue was isolated with TaKaRa MiniBEST Universal RNA Extraction Kit (TaKaRa, Ostu, Japan), and cDNA was generated using the iScript cDNA synthesis kit (Bio-Rad, Hercules, CA, USA) as per the manufacturers’ instructions. Real-time PCR was run on the QuantStudio™3 Real Time PCR System (Thermo Fisher Scientific) using the PowerUp™ SYBR™ Green Master Mix (Applied Biosystems, Foster City, CA, USA). The primer pairs were as follows: occludin: forward, 

5′-TTGAAAGTCCACCTCCTTACAGA-3′ and reverse, 5′-CCGGATAAAAAGAGTACGCTGG-3′; zonula occludens-1 (ZO-1): forward, 5′-GCCGCTAAGAGCACAGCAA-3′ and reverse, 5′-TCCCCACTCTGAAAATGAGGA-3′; TNF-α: forward, 5′-CCCTCACACTCAGATCATCTTCT-3′ and reverse, 5′-GCTACGACGTGGGCTACAG-3′; IL-1β: forward, 5′-GAAATGCCACCTTTTGACAGTG-3′ and reverse, 5′-TGGATGCTCTCATCAGGACAG-3′; IL-6: forward, 5′-TAGTCCTTCCTACCCCAATTTCC-3′ and reverse, 5′-TTGGTCCTTAGCCACTCCTTC-3′; GAPDH: forward, 5′-AGGTCGGTGTGAACGGATTTG-3′ and reverse, 5′-TGTAGACCATGTAGTTGAGGTCA-3′. 

GAPDH was used to normalize mRNA expression, and the quantification of gene expression was obtained using the 2^−ΔΔCT^ method.

### 2.8. Intestinal Permeability Analysis

Six-hour-fasted mice were orally administered 500 mg/kg of FITC-dextran. Blood was collected from the tail vein at 2 and 5 h after administration, and plasma was separated using centrifugation. Fluorescence of plasma was detected with a microplate reader (Molecular Devices, Sunnyvale, CA, USA) at an excitation and emission wavelength of 485/535 nm. The FITC-dextran concentration was determined from a standard curve of FITC-dextran.

### 2.9. β-Glucuronidase Activity

At week 12, fresh fecal samples were obtained and immediately stored at −80°. β-glucuronidase activity was analyzed based on the rate of *p*-nitrophenol release, according to the method of Juśkiewicz et al. [19], and the enzymatic activity was determined using a standard curve of *p*-nitrophenol.

### 2.10. Histological Analysis

Oil Red O (ORO) and hematoxylin and eosin (H&E) staining were performed to evaluate fat deposition in liver and WAT tissues. The liver or WAT tissues were directly embedded in OCT compound (Sakura Finetek, Torrance, CA, USA) or fixed with 4% formaldehyde. For ORO staining, OCT-embedded samples were cut at 10 μm. Then, slides were stained with ORO and counter-stained with hematoxylin. For H&E staining, fixed liver or WAT tissues were embedded into paraffin. Then, the tissues were cut at 4 μm and stained with H&E. Slides were pictured using a Pannoramic 250 Flash III slide scanner (3DHistech, Ltd., Budapest, Hungary) and images were acquired using CaseViewer software (3DHistech, Ltd., Budapest, Hungary). The H&E-stained tissue was scored for steatosis (0, none; 1, <33%; 2, 33–66%; 3, >66%) according to the method of Kleiner et al. [20]. Adipocyte area and ORO-positive staining areas were calculated using Image J software (NIH, Bethesda, MD, USA).

### 2.11. Statistical Analysis

Values are reported as means ± standard errors of the means. The data were analyzed using one-way analysis of variance followed by post hoc analysis using Tukey’s tests and a *p*-value < 0.05 was considered significant. All data were obtained using IBM SPSS Statistics 20 software (IBM Inc, Chicago, IL, USA).

## 3. Results

### 3.1. Effects of MG741 on Body Weight and Morphology of Adipose Tissue

Over the course of 12 weeks, the body weight and body fat percentages of HFD-fed mice were significantly increased without a change in food intake (Figure 1A–E). Moreover, adipocyte size in the HFD group also significantly increased compared with that in the ND group (Figure 1F). The 10^5^ CFU MG741 group did not exhibit significantly reduced weight gain, body fat percentage, or adipocyte size; however, the 10^6^ CFU MG741-administered group showed significantly lower values of these parameters compared with those in the HFD group.

### 3.2. Effects of MG741 on Metabolic Parameters

To confirm the effects of MG741 on metabolic disorder markers, we measured fasting blood glucose levels, serum levels of insulin, ALT, AST, leptin, total cholesterol, LDL-cholesterol, HDL-cholesterol, liver weight, and liver triglyceride levels. The HFD group had significantly increased fasting blood glucose and insulin levels, thereby inducing fasting hyperinsulinemia (Figure 2A–C). Moreover, the HFD group showed a significant increase in serum ALT, AST, leptin, total cholesterol, LDL-cholesterol, and HDL-cholesterol levels. The 10^5^ CFU MG741 administered group did not show improvements in any of the markers. MG741 did not change HDL-cholesterol levels but markedly reduced other metabolic disorder marker levels (Figure 2D–I). Increased body weight in the HFD-fed mice caused an increase in the liver weight and liver triglyceride level, but supplementation with a high dose of MG741 resulted in a decrease in these values relative to those in HFD mice (Figure 1F,G).

### 3.3. Effects of MG741 on Liver Lipid Metabolism

To determine if MG741 treatment could ameliorate fat deposition in the liver, lipid metabolism-related protein expression was assessed and histological analysis was performed. The HFD group exhibited significantly increased SREBP1, ChREBP, ACC, and FAS protein expression levels, by 2.5-, 2.3-, 2.9-, and 14.4-fold, respectively, compared with those in the ND group (Figure 3A). However, in the 10^6^ CFU MG741 group, levels of these proteins were significantly reduced by 57.8, 60.3, 66.5, and 29.5%, respectively, compared with those in the HFD group.

H&E staining showed a significant increase in fat deposition and steatosis scores in the liver (Figure 3B). ORO-positive staining areas were also markedly increased (Figure 3C). However, a high dose of MG741 significantly reduced liver fat deposition and the ORO-positive staining area.

### 3.4. Effects of MG741 on Gut Health

In the gut permeability analysis, the FITC-dextran level in plasma and the area under the curve were markedly increased in the HFD-fed mice (Figure 4A,B). Moreover, serum endotoxin levels and β-glucuronidase activity in the HFD group were drastically increased compared with those in the ND group (Figure 4C,D). The 10^5^ CFU MG741 group showed reduced β-glucuronidase activity, but the intestinal permeability and serum endotoxin levels were not significantly reduced. In contrast, the 10^6^ CFU MG741 group showed significant improvement in gut permeability and similar levels of endotoxin and β-glucuronidase activity compared with those in the ND group. In accordance with these results, the HFD group showed a reduction in levels of tight-junction-associated ZO-1 and occludin, and an increase in the levels of genes encoding inflammatory cytokines, such as TNF-α, IL-1β, and IL-6 (Figure 4E–I). However, these trends in gene expression were reversed in the high-dose MG741 treatment group, and the differences were statistically significant.

## 4. Discussion

In obese people, various metabolic disorders, such as insulin resistance, inflammation, and NAFLD, can occur. Several studies have shown that metabolic disorders are caused by a disruption to the gut barrier function due to poor eating habits [21]. Numerous probiotics are known to suppress metabolic disorders by improving the gut barrier dysfunction [22]. Therefore, we determined if beneficial bacteria (MG741) obtained from the feces of a healthy breast-fed infant could ameliorate body weight gains and other metabolic disorders by improving gut health. In this study, as expected, HFD-fed mice showed significantly increased body weight and fat deposition. In addition, it was confirmed that various NAFLD-related indicators were also increased. Among them, obesity-induced insulin resistance and increased leptin levels are known to be among the main causes of NAFLD [23]. Increased levels of insulin upregulate leptin, thereby inducing leptin resistance in the nervous system and adipose tissue [24]. Therefore, leptin does not reduce insulin levels and promotes hepatic lipid accumulation. A high dose of MG741 treatment not only reduced the body weight gain and fat deposition induced by an HFD but also improved fasting hyperinsulinemia and decreased serum leptin levels. In addition, 10^6^ CFU MG741 improved levels of ALT and AST, known as markers of liver injury and surrogate measures of NAFLD, and reduced levels of liver triglycerides. These results suggest that a high dose of MG741 might ameliorate HFD-induced metabolic disorders.

To determine whether changes in blood markers support the occurrence of NAFLD mediated by an HFD, lipid metabolism-related protein expression was assessed and histological analysis was performed. An HFD significantly increased the expression of SREBP-1 and ChREBP, and also markedly upregulated the expression of their downstream signals, namely ACC and FAS. SREBP1 and ChREBP are well-known key regulators of lipid metabolism. SREBP1 transcriptionally activates genes involved in fatty acid synthesis and is activated by insulin, whereas ChREBP activates both glycolysis and fatty acid synthesis and is activated by glucose. In addition, SREBP1 and ChREBP directly regulate each other to activate lipogenic and glycolytic genes, thereby inducing hepatic lipogenesis [25]. ACC and FAS are representative triglyceride synthesis genes, and many studies have reported that they are regulated by the combined action of SREBP1 and ChREBP [26,27]. In this study, it was also confirmed that the levels of liver triglycerides and hepatic fat deposition were significantly increased due to these HFD-induced protein changes. These lipid metabolism and histological changes were significantly reduced with MG741 supplementation. Taken together, we suggest that MG741 can inhibit NAFLD by changing not only blood markers but also lipid metabolism-related proteins.

We tried to determine if MG741 directly affected the liver or whether it affected the liver through the gut–liver axis. An HFD is known to induce gut dysbiosis, resulting in the production of various toxic substances, such as LPS, and leading to metabolic endotoxemia [28]. In addition, harmful bacteria in the gut deteriorate intestinal health by producing a harmful enzyme, β-glucuronidase [29]. β-Glucuronidase catalytically hydrolyzes glucuronides, and these metabolites cause damage to the gut [30]. Increased gut harmful bacteria and toxic substances are known to increase gut permeability by causing inflammation and damaging tight junctions, thereby leading to metabolic endotoxemia [31]. Through this process, body weight gain, insulin resistance, and de novo fatty acid synthesis can occur, which eventually leads to NAFLD [32]. Our results confirm previously reported observations that gut permeability was significantly increased by HFD, and the concentration of serum endotoxin was also increased. In addition, through the increased β-glucuronidase activity, it could be assumed that the cause of this phenomenon might be due to the increase in toxic substances caused by disruption of the gut microbiota. ZO-1 and occludin are key markers of tight junction integrity [33]. Inflammation in the gut triggers disruption of the tight junction [34], and therefore, reductions in ZO-1 and occludin mRNA expression are responsible for increased gut permeability. In this study, increased gene expression of inflammatory cytokines and decreased gene expression of ZO-1 and occludin were confirmed in HFD-fed mouse colons. Taken together, we speculated that toxic substances derived from deteriorated gut health might have had an effect on the occurrence of NAFLD through the gut–liver axis. Supplementation with 10^6^ CFU MG741 was shown to improve disrupted gut barrier functions, including inflammation and, in particular, it significantly increased tight junction-related gene expression. These results suggest that MG741 suppresses weight gain and NAFLD caused by an HFD by improving gut health. Supplementation with 10^5^ CFU MG741 reduced β-glucuronidase activity as much as 10^6^ CFU MG741, but did not improve changes in various gut–liver-axis-related markers. This was thought to be because the serum levels of endotoxin, which plays a major role in gut permeability and the development of NAFLD, could not be dramatically improved. Therefore, we speculate that endotoxin is important in the development of NAFLD and is an essential marker for the improvement of the gut–liver axis.

The present study showed that MG741, a bacterium isolated from a healthy infant, can ameliorate the expression of inflammatory cytokines in the liver and increase tight junction gene expression in the colon. These changes were determined to be associated with the attenuation of NAFLD, weight loss, and a reduction in liver and gut inflammation in an obese mouse model. Taken together, supplementation with MG741 might be a new, effective, and safe preventive dietary strategy for NAFLD.

## Figures and Tables

**Figure 1 nutrients-14-01965-f001:**
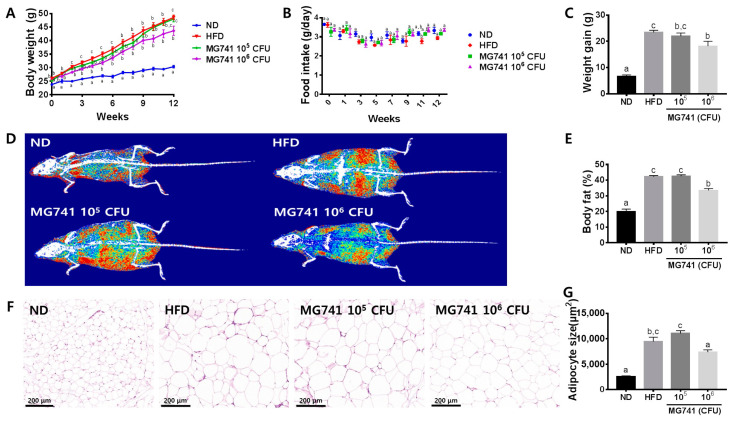
*Bifidobacterium animalis* ssp. *lactis* MG741 (MG741) ameliorates high-fat-diet-induced fat deposition. (**A**) Changes in body weight during the experimental period; (**B**) changes in food intake during the experimental period; (**C**) weight gain; (**D**) representative body composition images; (**E**) body fat percent; and (**F**) representative histological results based on H&E staining in white adipose tissue and quantification of adipocyte size. (**G**) Adipocyte size. Data are presented as the mean ± standard error of the mean (*n* = 9). The different letters (a–c) indicate significant differences (*p* < 0.05) determined by one-way ANOVA with Tukey’s post hoc tests.

**Figure 2 nutrients-14-01965-f002:**
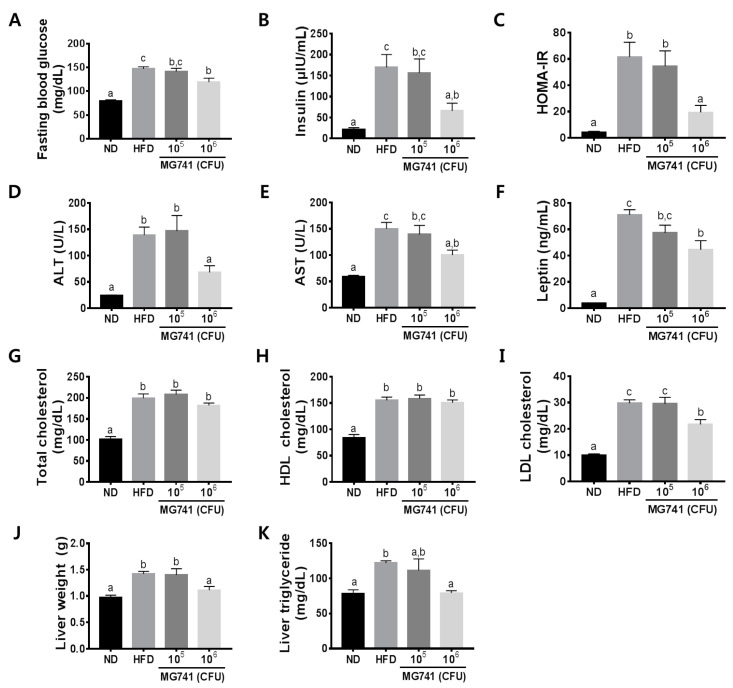
*Bifidobacterium animalis* ssp. *lactis* MG741 (MG741) ameliorates high-fat-diet-induced metabolic disorder parameters. (**A**) Fasting blood glucose; (**B**) insulin; (**C**) HOMA-IR; (**D**) serum ALT; (**E**) serum AST; (**F**) serum leptin; (**G**) serum total cholesterol; (**H**) serum HDL-cholesterol; (**I**) serum LDL-cholesterol; (**J**) liver weight; and (**K**) liver triglycerides. Data are presented as the mean ± standard error of the mean (*n* = 9). The different letters (a–c) indicate significant differences (*p* < 0.05) determined by one-way ANOVA with Tukey’s post hoc tests.

**Figure 3 nutrients-14-01965-f003:**
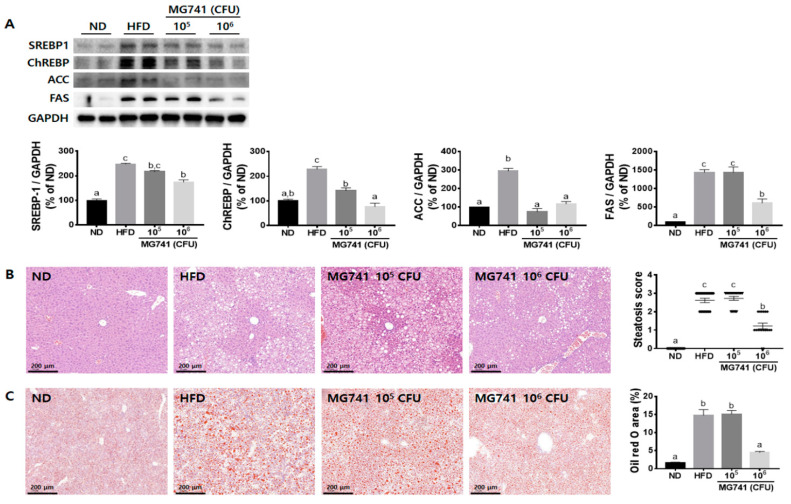
*Bifidobacterium animalis* ssp. *lactis* MG741 (MG741) ameliorates high-fat-diet-induced changes in lipid metabolism in the liver. (**A**) Representative Western blots of SREBP-1, ChREBP, ACC, and FAS. (**B**) Representative histological results based on H&E staining and steatosis score. (**C**) Representative histological results based on Oil Red O staining and the positive staining area. Data are presented as the mean ± standard error of the mean (*n* = 6). The different letters (a–c) indicate significant differences (*p* < 0.05) determined by one-way ANOVA with Tukey’s post hoc tests.

**Figure 4 nutrients-14-01965-f004:**
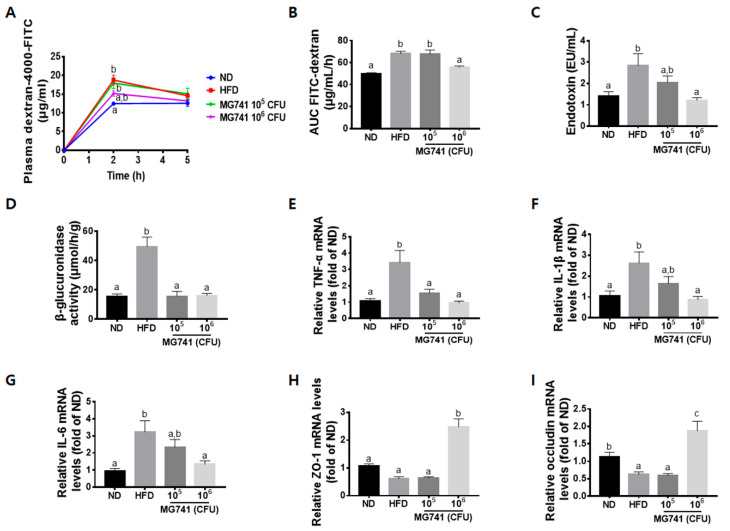
*Bifidobacterium animalis* ssp. *lactis* MG741 (MG741) improves high-fat-diet-induced changes in gut barrier function. (**A**) Plasma FITC-dextran concentration during the gut permeability test; (**B**) area under the curve of the plasma FITC-dextran levels; (**C**) plasma endotoxin; (**D**) β-glucuronidase activity; (**E**) relative mRNA expression of TNF-α; (**F**) relative mRNA expression of IL-1β; (**G**) relative mRNA expression of IL-6; (**H**) relative mRNA expression of ZO-1; and (**I**) relative mRNA expression of occludin. Data are presented as the mean ± standard error of the mean (*n* = 9). The different letters (a–c) indicate significant differences (*p* < 0.05) determined by one-way ANOVA with Tukey’s post hoc tests.

## Data Availability

Not applicable.

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
