# Peer review of "Bifidobacterium animalis ssp. lactis MG741 Reduces Body Weight and Ameliorates Nonalcoholic Fatty Liver Disease via Improving the Gut Permeability and Amelioration of Inflammatory Cytokines"

_nutrients, 2022, doi:10.3390/nu14091965_

Round 1
Reviewer 1 Report
The paper “Bifidobacterium animalis ssp. lactis MG741 reduces body 2 weight and ameliorates nonalcoholic fatty liver disease via the 3 gut-liver axis" by Moon Ho Do et al. deals with a highly important subject study.
The article is well written and only minor spell check is necessary. The paper has a good design. The article is logically divided into sections and subsections. The work has a good degree of novelty and of good interest to the readers.
Comments:
- Line 32: NAFLD development does not only derives from lipid accumulation, but also by the increased ROS production due to increased mitochondria activity, which in turn worsen the hepatic insulin resistance. (doi: https://doi.org/10.31083/j.rcm2203082)
- Line 36: The mechanism of liver damage due to endotoxemia is linked to the fact that “over 50% of the blood supply in the liver is provided by the splanchnic district, thus being among the organs most exposed to toxins of intestinal origin, as it represents the first line of defence against products of bacterial origin. The crosstalk between host and microbiota along the intestinal mucosal interface is fundamental for both the development and homeostasis of the host’s innate and adaptive immune system” (doi: 10.3390/pr9010135)
Author Response
We highly appreciate the reviewer’s constructive and helpful comments on our manuscript. As suggested by the reviewer, we have carefully response (marked in blue) to address the reviewer’s comments and revised manuscript (marked in red). We hope that the reviewer will find our responses to the comments satisfactory.
Reviewer 1:
The paper “Bifidobacterium animalis ssp. lactis MG741 reduces body 2 weight and ameliorates nonalcoholic fatty liver disease via the 3 gut-liver axis" by Moon Ho Do et al. deals with a highly important subject study.
The article is well written and only minor spell check is necessary. The paper has a good design. The article is logically divided into sections and subsections. The work has a good degree of novelty and of good interest to the readers.
Comments:
Line 32: NAFLD development does not only derives from lipid accumulation, but also by the increased ROS production due to increased mitochondria activity, which in turn worsen the hepatic insulin resistance. (doi: https://doi.org/10.31083/j.rcm2203082)
▶ We appreciated reviewer’s kind comment. According to the comment, we revised introduction and added the reference.
Lines 32–35: Obesity causes hepatic insulin resistance by increasing mitochondrial reactive oxygen species production and the release of free fatty acids, which leads to the accumulation of fatty acids in the liver, thereby inducing NAFLD [6-8].
Reference: 8. Galiero, R.; Caturano, A.; Vetrano, E.; Cesaro, A.; Rinaldi, L.; Salvatore, T.; Marfella, R.; Sardu, C.; Moscarella, E.; Gragnano, F. Pathophysiological mechanisms and clinical evidence of relationship between Nonalcoholic fatty liver disease (NAFLD) and cardiovascular disease. Reviews in cardiovascular medicine 2021, 22, 755-768.
Line 36: The mechanism of liver damage due to endotoxemia is linked to the fact that “over 50% of the blood supply in the liver is provided by the splanchnic district, thus being among the organs most exposed to toxins of intestinal origin, as it represents the first line of defence against products of bacterial origin. The crosstalk between host and microbiota along the intestinal mucosal interface is fundamental for both the development and homeostasis of the host’s innate and adaptive immune system” (doi: 10.3390/pr9010135)
▶ According to the opinion of the reviewer, we revised introduction and added the reference.
Lines 39–41: Since the liver is provided more than 50% of blood by the splanchnic district, it is one of the organs most exposed to gut-derived toxins [11]. Therefore, increased endotoxin in circulating blood causes insulin resistance, inflammation, and fat accumulation, eventually progressing to NAFLD [12].
Reference: 11. Caturano, A.; Acierno, C.; Nevola, R.; Pafundi, P.C.; Galiero, R.; Rinaldi, L.; Salvatore, T.; Adinolfi, L.E.; Sasso, F.C. Non-alcoholic fatty liver disease: from pathogenesis to clinical impact. Processes 2021, 9, 135.
※ We would like to thank the reviewers for the constructive and insightful comments and hope that our answers will be acceptable to the reviewers.

Reviewer 2 Report
In the present article, Do et al., have studied the effects of a probiotic, Bifidobacterium animalis ssp on liver metabolism in a context of HFD-induced obesity and nonalcoholic fatty liver disease. They showed that this bacterium improved all the features of HFD feeding including obesity, hepatic steatosis, adiposity, inflammation and gut permeability. This article is informative and fits well with the topic of the journal. However, some critical points need to be improved. Major and minor comments are listed below.
Major issues:
Authors need to measure the fecal or cecal abundance of Bifidobacterium animalis ssp to verify if the experiment worked.
In addition to western blot analysis, authors should measure the RNA expression of lipogenic markers (SREBP1, ChREBP, ACC, FAS, GAPDH).
Page 4 line 143-144, authors wrote “The H&E-stained tissue was scored for steatosis according to the method of Kato et al.,” but the scoring of hepatic steatosis comes from Kleiner et al., in 2005 (PMID: 15915461) and Kato et al., have already used this method of scoring. Please modify the sentence with the correct reference.
Page 3, line 92-93, authors wrote “insulin and endotoxin levels were determined with a kit from Thermo Fisher Scientific”. Authors need to provide more information about the way of quantifying endotoxin and the reference of the kits.
Did the authors perform an OGTT to assess glucose tolerance? Page 7, line 236-237, authors talked about insulin resistance but they didn’t measure it properly. Authors should replace insulin resistance by “fasting hyperinsulinemia”.
In the discussion part, the authors explain a potential effect of the bacterium on beta-glucuronidase, which would therefore be responsible for the beneficial effects on NAFLD. How do you explain that low dose (10^5) of the bacterium prevented HFD-induced increase b-glucuronidase but had no effect on metabolism? Is there an alternative mechanism that explain the protection against NAFLD? Authors are encouraged to go further in the discussion of the results.
Minor issues:
Page 1, line 42: “Bifidobacteria are considered one of the key microbiota”. This sentence is unclear and should be rewritten
Figure 1 and 3: Scale for histological pictures is missing
Author Response
We highly appreciate the reviewer’s constructive and helpful comments on our manuscript. As suggested by the reviewer, we have carefully response (marked in blue) to address the reviewer’s comments and revised manuscript (marked in red). We hope that the reviewer will find our responses to the comments satisfactory.
Reviewer 2:
In the present article, Do et al., have studied the effects of a probiotic, Bifidobacterium animalis ssp on liver metabolism in a context of HFD-induced obesity and nonalcoholic fatty liver disease. They showed that this bacterium improved all the features of HFD feeding including obesity, hepatic steatosis, adiposity, inflammation and gut permeability. This article is informative and fits well with the topic of the journal. However, some critical points need to be improved. Major and minor comments are listed below.
Major issues:
Authors need to measure the fecal or cecal abundance of Bifidobacterium animalis ssp to verify if the experiment worked.
▶ Regardless of the presence or absence of intestinal colonization of probiotics, the ingested probiotics are measured during NGS analysis using mouse feces after ingestion of probiotics. In addition, when performing gut-liver axis studies, it is reasonable to perform NGS analysis, but in this study, we unfortunately did not perform NGS analysis in this study. In the further study, we will confirm whether this strain settles well in the gut and its proportion is increased through NGS analysis.
In addition to western blot analysis, authors should measure the RNA expression of lipogenic markers (SREBP1, ChREBP, ACC, FAS, GAPDH).
▶ Many papers show RNA expression to determine changes of lipogenic markers, but there are also many papers that show changes of those markers in protein expression. In addition, since the patterns of RNA expression and protein expression were similar in several papers (1), it can be determined that the expression of lipogenic markers in the liver was also changed by western blot results. In the further study, we will analyze both RNA expression and protein expression according to the reviewer's opinion.
(1) Wang, Z.; Li, B.; Jiang, H.; Ma, Y.; Bao, Y.; Zhu, X.; Xia, H.; Jin, Y. IL-8 exacerbates alcohol-induced fatty liver disease via the Akt/HIF-1α pathway in human IL-8-expressing mice. Cytokine 2021, 138, 155402.
Page 4 line 143-144, authors wrote “The H&E-stained tissue was scored for steatosis according to the method of Kato et al.,” but the scoring of hepatic steatosis comes from Kleiner et al., in 2005 (PMID: 15915461) and Kato et al., have already used this method of scoring. Please modify the sentence with the correct reference.
▶ According to the opinion of the reviewer, we revised reference.
Lines 150–152: The H&E-stained tissue was scored for steatosis (0, none; 1, <33%; 2, 33–66%; 3, >66%) ac-cording to the method of Kleiner et al. [20].
Page 3, line 92-93, authors wrote “insulin and endotoxin levels were determined with a kit from Thermo Fisher Scientific”. Authors need to provide more information about the way of quantifying endotoxin and the reference of the kits.
▶ For endotoxin determination, Pierce™ Chromogenic Endotoxin Quant Kit (A39553) was used. This kit is a very popular kit used to measure blood endotoxin, so we did not wrote information about it. We wrote commercial kit information in 2.5. Biochemical analysis.
Lines 90–102: At the end of the experiment, mice were fasted for 12 h and euthanized by anesthesia. Then, tissues were collected and immediately stored at −80°C until use. Blood was col-lected from the abdominal vein and serum was separated using centrifugation. The ob-tained serum was stored at −80°C until further processing. Serum alanine aminotransfer-ase (ALT) and aspartate aminotransferase (AST) contents were estimated using kits from Cusabio (CSB-E16539m, CSB-E12649m, Wuhan, China). Liver triglyceride and serum total cholesterol, high-density lipoprotein (HDL) cholesterol and low-density lipoprotein (LDL) cholesterol were measured with kits from Abcam (ab65336 and ab65390). Serum leptin was determined using kits from R&D systems (MOB00B, Minneapolis, MN, USA), and insulin and endotoxin levels were determined with a kit from Thermo Fisher Scientific (EMINS, A39553). All data were quantified as per the manufacturers’ instructions and insulin resistance was calculated as HOMA-IR = fasting insulin (mU/L) × fasting glucose (mmol/L)/22.5 [16].
Did the authors perform an OGTT to assess glucose tolerance? Page 7, line 236-237, authors talked about insulin resistance but they didn’t measure it properly. Authors should replace insulin resistance by “fasting hyperinsulinemia”.
▶ Although OGTT was not performed, it was determined that insulin resistance increased as a result of HOMA-IR measurement. However, Thomas et al. reported that in subjects with obesity but without diabetes or hypertension, hyperinsulinemia and insulin hypersecretion are more prevalent than insulin resistance and hence may precede and contribute to insulin resistance (2). Therefore, it was judged appropriate to change to “fasting hyperinsulinemia” according to the opinion of the reviewer, and “insulin resistance” written in the manuscript was changed to “fasting hyperinsulinemia”.
(2) Thomas, D.D.; Corkey, B.E.; Istfan, N.W.; Apovian, C.M. Hyperinsulinemia: an early indicator of metabolic dysfunction. Journal of the Endocrine Society 2019, 3, 1727-1747.
Lines 15–17: HFD-fed mice were supplemented daily with MG741. After 12 weeks, MG741-administered mice exhibited reduced fat deposition, and serum metabolic alterations, including fasting hyperinsulinemia, were modulated.
Lines 25–26: Keywords: Bifidobacterium animalis ssp. lactis MG741; non-alcoholic fatty liver disease; fasting hyperinsulinemia; gut health; gut-liver axis
Lines 178–179: The HFD group had significantly increased fasting blood glucose and insulin levels, thereby inducing fasting hyperinsulinemia (Figure 2A–C).
Lines 246–248: A high dose of MG741 treatment not only reduced the body weight gain and fat deposition induced by HFD but also improved fasting hyperinsulinemia and decreased serum leptin levels.
In the discussion part, the authors explain a potential effect of the bacterium on beta-glucuronidase, which would therefore be responsible for the beneficial effects on NAFLD. How do you explain that low dose (10^5) of the bacterium prevented HFD-induced increase b-glucuronidase but had no effect on metabolism? Is there an alternative mechanism that explain the protection against NAFLD? Authors are encouraged to go further in the discussion of the results.
▶ Low-dose MG741 dramatically decreased the β-glucuronidase activity, but did not significantly reduce serum endotoxin levels. Improvement of endotoxin is also very important for improvement of Gut-liver axis (3). Therefore, it is thought that various markers could not be improved because the serum levels of endotoxin could not be reduced as much as the high dose at the low dose. We wrote about this in the discussion.
(3) Nagata, N.; Xu, L.; Kohno, S.; Ushida, Y.; Aoki, Y.; Umeda, R.; Fuke, N.; Zhuge, F.; Ni, Y.; Nagashimada, M. Glucoraphanin ameliorates obesity and insulin resistance through adipose tissue browning and reduction of metabolic endotoxemia in mice. Diabetes 2017, 66, 1222-1236.
Lines 292–297: Low-dose (105 CFU/mouse) MG741 also reduced β-glucuronidase activity as much as high-dose (106 CFU/mouse) MG741, but did not improve changes of various gut-liver axis-related markers. This is thought to be because the serum levels of endotoxin, which plays a major role in gut permeability and the development of NAFLD, could not be dramatically improved. Therefore, it is speculated that endotoxin is important in the development of NAFLD and implies that it is an essential marker for improvement of the gut-liver axis.
Minor issues:
Page 1, line 42: “Bifidobacteria are considered one of the key microbiota”. This sentence is unclear and should be rewritten
▶ We rewrote this sentence.
Lines 46–47: Bifidobacteria are considered one of the important health-beneficial bacteria in the intestinal tracts, with a high abundance within the newborn gut during breastfeeding [14].
Figure 1 and 3: Scale for histological pictures is missing
▶ Scale bars were added according to the reviewer's comment.
※ We would like to thank the reviewers for the constructive and insightful comments and hope that our answers will be acceptable to the reviewers.

Reviewer 3 Report
The manuscript was reviewed for publication in the journal. The manuscript was designed to evaluate the effects of Bifidobacterium animalis ssp. lactis MG741 on nonalcoholic fatty liver disease (NAFLD) and weight gain using a mouse model of high-fat diet (HFD)-induced obesity. The results obtained indicate that the oral administration of MG741 could prevent NAFLD and obesity, thereby improving metabolic health. It is the reviewer’s opinion that the manuscript is interesting and easy to follow. However, it appears that there are a couple of concerns in the manuscript.
1) A dose of MG741 was administered at a dose of 1.0 × 105 or 1.0 × 106 CFU/day/mouse in the experiment. How about other dose (1.0 × 107)? Is there any reason to use these doses? The authors should explain the point.
2) Figure 1D showed representative body composition images. The authors should explain the meaning of color.
3) The authors should add a bar scale in Figure 1F and Figure 3B and C.
4) Figure 3B showed representative histological results based on HE staining and steatosis score. The authors should use a dot plot in the steatosis score.
5) The authors mentioned that MG741 improved the HFD-induced deterioration in gut permeability by reducing toxic substances and inflammatory cytokine expression and upregulating tight junction. How about toxic substances? The authors should explain how to improve gut permeability by reducing inflammatory cytokine expression and upregulating tight junctions.
6) The authors should show the protein expression of tight junctions instead of relative mRNA expression.
Author Response
We highly appreciate the reviewer’s constructive and helpful comments on our manuscript. As suggested by the reviewer, we have carefully response (marked in blue) to address the reviewer’s comments and revised manuscript (marked in red). We hope that the reviewer will find our responses to the comments satisfactory.
Reviewer 3:
The manuscript was reviewed for publication in the journal. The manuscript was designed to evaluate the effects of Bifidobacterium animalis ssp. lactis MG741 on nonalcoholic fatty liver disease (NAFLD) and weight gain using a mouse model of high-fat diet (HFD)-induced obesity. The results obtained indicate that the oral administration of MG741 could prevent NAFLD and obesity, thereby improving metabolic health. It is the reviewer’s opinion that the manuscript is interesting and easy to follow. However, it appears that there are a couple of concerns in the manuscript.
1) A dose of MG741 was administered at a dose of 1.0 × 105 or 1.0 × 106 CFU/day/mouse in the experiment. How about other dose (1.0 × 107)? Is there any reason to use these doses? The authors should explain the point.
▶ After confirming the efficacy at the low dose (1.0 × 105 or 1.0 × 106), we were going to assess whether the high dose shows better efficacy without side effects. However, since sufficient functionality of MG741 was confirmed at both concentrations of 1.0 × 105 or 1.0 × 106, concentrations higher than that were not evaluated.
2) Figure 1D showed representative body composition images. The authors should explain the meaning of color.
▶ Radiography of body fat was displayed by three modes according to low density fat (blue color), medium density fat (yellow color) and high density fat (red color). We wrote about this in the 2.4. InAlyzer analysis.
Lines 85–87: The body composition was revealed by three modes into lean tissue (blue color), medium density fat (yellow color) and high density fat (red color).
3) The authors should add a bar scale in Figure 1F and Figure 3B and C.
▶ Scale bars were added according to the reviewer's opinion.
Lines 167 and 206:
4) Figure 3B showed representative histological results based on HE staining and steatosis score. The authors should use a dot plot in the steatosis score.
▶ The graph was changed according to the reviewer's opinion.
Lines 206:
5) The authors mentioned that MG741 improved the HFD-induced deterioration in gut permeability by reducing toxic substances and inflammatory cytokine expression and upregulating tight junction. How about toxic substances? The authors should explain how to improve gut permeability by reducing inflammatory cytokine expression and upregulating tight junctions.
▶ Tian et al. reported that Metabolic endotoxins are also related to elevated TNF-α and IL-6 levels. Moreover, lipopolysaccharide (LPS) also increases gut permeability and inflammation via the toll-like receptor 4 (TLR4) signaling pathway (1). In addition, we also described in the discussion section (pages 8, 270-275): HFD is known to induce gut dysbiosis, resulting in the production of various toxic substances such as LPS and leading to metabolic endotoxemia [28]. In addition, harmful bacteria in the gut deteriorate intestinal health by producing a harmful enzyme, β-glucuronidase [29]. β-Glucuronidase catalytically hydrolyzes glucuronides, and these metabolites cause damage to the gut [30]. Increased gut harmful bacteria and toxic substances are known to increase gut permeability by causing inflammation and damaging tight junctions, thereby leading to metabolic endotoxemia [31].
In this study, 106 CFU MG741 showed decrease of production of toxic substances such as endotoxin (LPS) and β-glucuronidase. These reduced intestinal toxic substances are thought to have a positive effect on the gut barrier function. Therefore, it is considered that toxic substances such as endotoxin are decreased due to supplementation with MG741, which leads to decreased expression of inflammatory cytokines and increased expression of ZO-1 and occludin. We added sentences to the discussion.
(1) Tian, B.; Zhao, J.; Zhang, M.; Chen, Z.; Ma, Q.; Liu, H.; Nie, C.; Zhang, Z.; An, W.; Li, J. Lycium ruthenicum Anthocyanins Attenuate High‐Fat Diet‐Induced Colonic Barrier Dysfunction and Inflammation in Mice by Modulating the Gut Microbiota. Molecular nutrition & food research 2021, 65, 2000745.
Lines 292–297: Low-dose (105 CFU/mouse) MG741 also reduced β-glucuronidase activity as much as high-dose (106 CFU/mouse) MG741, but did not improve changes of various gut-liver axis-related markers. This is thought to be because the serum levels of endotoxin, which plays a major role in gut permeability and the development of NAFLD, could not be dramatically improved. Therefore, it is speculated that endotoxin is important in the development of NAFLD and implies that it is an essential marker for improvement of the gut-liver axis.
6) The authors should show the protein expression of tight junctions instead of relative mRNA expression.
▶ Many papers show RNA expression to determine changes of tight junctions (2). In addition, since the patterns of RNA expression and protein expression were similar in several papers (3), it can be determined that the expression of tight junctions in the colon was also changed by real-time PCR results. In the further study, we will analyze both RNA expression and protein expression according to the reviewer's opinion.
(2) Oh, Y.J.; Kim, H.J.; Kim, T.S.; Yeo, I.H.; Ji, G.E. Effects of Lactobacillus plantarum PMO 08 alone and combined with chia seeds on metabolic syndrome and parameters related to gut health in high-fat diet-induced obese mice. Journal of Medicinal Food 2019, 22, 1199-1207.
(3) Liu, S.; Yu, H.; Li, P.; Wang, C.; Liu, G.; Zhang, X.; Zhang, C.; Qi, M.; Ji, H. Dietary nano-selenium alleviated intestinal damage of juvenile grass carp (Ctenopharyngodon idella) induced by high-fat diet: Insight from intestinal morphology, tight junction, inflammation, anti-oxidization and intestinal microbiota. Animal Nutrition 2022, 8, 235-248.
※ We would like to thank the reviewers for the constructive and insightful comments and hope that our answers will be acceptable to the reviewers.

Reviewer 4 Report
This manuscript is well presented and the originality is good. However, the results in this work only briefly explain the amelioration of MG741 in NAFLD by gut-liver axis, probably due to anti-inflammatory effects and gut barrier improvement. The author should add more results, such as the expression levels of lipid metabolism markers in visceral fat and the experiment of glucose tolerance test, to provide mechanistic evidence why MG741 could reduce body weight in obese mice. Major revision is needed.
Major:
- Abstract
The language in this part should be polished. Also, more detailed information should be added. For examples, the sentence “MG741 regulated lipid metabolism and lipid accumulation in the liver” in line 17 should include which parameters were determined to derive this statement.
- Methods:
Is 12 weeks long enough to induce NAFLD using high fat diet? Why did the author select these two doses of MG741 in line 75? Is there any dosage regimen for the dosage being used? What is the difference between B. animalis ssp. Lactis MG741 and other Bifidobacterial strains on body weight gain, or could other Bifidobacterial strains reduce body weight and improve NAFLD? Was there another Bifidobacterium spp. or another B. animalis strain tested to see if the anti-obesity effect is uniquely present for MG741?
(3) Results:
The author may need to add ITT and GTT experiments, measuring the expression levels of lipid metabolism markers in visceral fat, and the results of gut microbiota, to complete the whole story of this manuscript.
Minor:
Line 13: “regulate” should be changed into “improve”.
Line 18: “In determine” should be changed into “To determine”.
Line 47: “has strong antioxidant activities” should be changed into “exhibited strong antioxidant activities”.
Author Response
We highly appreciate the reviewer’s constructive and helpful comments on our manuscript. As suggested by the reviewer, we have carefully response (marked in blue) to address the reviewer’s comments and revised manuscript (marked in red). We hope that the reviewer will find our responses to the comments satisfactory.
Reviewer 4:
This manuscript is well presented and the originality is good. However, the results in this work only briefly explain the amelioration of MG741 in NAFLD by gut-liver axis, probably due to anti-inflammatory effects and gut barrier improvement. The author should add more results, such as the expression levels of lipid metabolism markers in visceral fat and the experiment of glucose tolerance test, to provide mechanistic evidence why MG741 could reduce body weight in obese mice. Major revision is needed.
Major:
(1) Abstract
The language in this part should be polished. Also, more detailed information should be added. For examples, the sentence “MG741 regulated lipid metabolism and lipid accumulation in the liver” in line 17 should include which parameters were determined to derive this statement.
▶ According to the reviewer’s kind comment, we revised abstract.
Lines 17–20: In addition, MG741 regulated Acetyl-CoA carboxylase, fatty acid synthase, sterol regulatory element-binding protein 1, and carbohydrate-responsive element-binding protein expressions and lipid accumulation in the liver, thereby reducing the hepatic steatosis score.
(2) Methods:
Is 12 weeks long enough to induce NAFLD using high fat diet? Why did the author select these two doses of MG741 in line 75? Is there any dosage regimen for the dosage being used? What is the difference between B. animalis ssp. Lactis MG741 and other Bifidobacterial strains on body weight gain, or could other Bifidobacterial strains reduce body weight and improve NAFLD? Was there another Bifidobacterium spp. or another B. animalis strain tested to see if the anti-obesity effect is uniquely present for MG741?
▶ In many papers, NAFLD research is conducted using a high fat diet, and there are many papers reporting that NAFLD appears even in a diet for 8 weeks (1). In addition, several researchers are conducting anti-obesity or protective effect on NAFLD studies using various B. animalis strains (2). Moreover, Wang et al. reported that B. bifidum and B. adolescentis also exert significant mitigative effects on NAFLD caused by high-fat diets (3). Therefore, anti-obesity and/or NAFLD improvement ability may differ slightly depending on the Bifidobacterial strain, but it is thought to be a common efficacy.
After confirming the efficacy at the low dose, we were going to check whether the high dose shows better efficacy without side effects. Therefore, the study was conducted first at 1.0 × 105 or 1.0 × 106 CFU. In the further study, we will evaluate whether high doses of MG741 have better effects (1.0 × 107 or 1.0 × 108).
(1) Shin, M.K.; Yang, S.-M.; Han, I.-S. Capsaicin suppresses liver fat accumulation in high-fat diet-induced NAFLD mice. Animal cells and systems 2020, 24, 214-219.
(2) Yan, Y.; Liu, C.; Zhao, S.; Wang, X.; Wang, J.; Zhang, H.; Wang, Y.; Zhao, G. Probiotic Bifidobacterium lactis V9 attenuates hepatic steatosis and inflammation in rats with non-alcoholic fatty liver disease. AMB Express 2020, 10, 1-11.
(3) Wang, L.; Jiao, T.; Yu, Q.; Wang, J.; Wang, L.; Wang, G.; Zhang, H.; Zhao, J.; Chen, W. Bifidobacterium bifidum Shows More Diversified Ways of Relieving Non-Alcoholic Fatty Liver Compared with Bifidobacterium adolescentis. Biomedicines 2021, 10, 84.
(3) Results:
The author may need to add ITT and GTT experiments, measuring the expression levels of lipid metabolism markers in visceral fat, and the results of gut microbiota, to complete the whole story of this manuscript.
▶ Unfortunately, we did not perform ITT, GTT and NGS analysis. Moreover, we just check the body fat mass and adipocyte expansion in epididymal adipose tissue. So, in the title of the paper, we wrote only “reduces body weight”, not the “improves obesity” However, we were able to confirm the occurrence of metabolic disorders by calculating HOMA-IR through measuring fasting blood glucose and insulin levels. In a further study using high dose of MG741, we will perform the above experiments and derive the results.
Minor:
Line 13: “regulate” should be changed into “improve”.
▶ We revised " regulate " to " improve."
Lines 12–13: Recently, specific probiotic strains have been found to improve symptoms of NAFLD.
Line 18: “In determine” should be changed into “To determine”.
▶ We revised "In determining" to "To determine."
Lines 20–23: To determine whether the effects of MG741 were related to improvements in gut health, MG741 improved the HFD-induced deterioration in gut permeability by reducing toxic substances and inflammatory cytokine expression and upregulating tight junctions.
Line 47: “has strong antioxidant activities” should be changed into “exhibited strong antioxidant activities”.
▶ We revised “has strong antioxidant activities” to “exhibited strong antioxidant activities”.
Lines 50–54: Our previous study reported that Bifidobacterium animalis ssp. lactis MG741 (MG741) isolated from human infant feces exhibited strong antioxidant activities such as nitric oxide, 2,2-diphenyl-1-picrylhydrazyl (DPPH) free radical-scavenging and 2,2′-azino-bis-3-ethylbenzothiazoline-6-sulfonic acid (ABTS) radical-scavenging [15].
※ We would like to thank the reviewers for the constructive and insightful comments and hope that our answers will be acceptable to the reviewers.

Round 2
Reviewer 2 Report
The authors answered to the remarks properly but they didn't permormed the experiments asked by the reviewer. Some points are critical and need to be solved before publishing the paper in this journal. The quantity of the bacteria needs to be measured to appreciate the viability of the experiment. And mRNA expression of lipogenic markers needs to be measured since these markers are highly regulated by transcription.
Author Response
Reviewer 2:
The authors answered to the remarks properly but they didn't permormed the experiments asked by the reviewer. Some points are critical and need to be solved before publishing the paper in this journal. The quantity of the bacteria needs to be measured to appreciate the viability of the experiment. And mRNA expression of lipogenic markers needs to be measured since these markers are highly regulated by transcription.
▶ We agree with the comments of reviewers and believe that demonstrating them will lead to better paper. However, we have been given two days by the editor to process major revision.
In fact, we provided fecal samples to the company for NGS analysis. Unfortunately, NGS analysis did not go well (samples problem or analysis equipment problem), so we could not get results worthy of being included in the paper.
Moreover, we lacked the reagents needed to conduct the experiment for gene expression during the revision period, and there was not enough time for analysis. Therefore, we checked various papers and confirmed that changes in lipogenic markers can be judged by protein expression (1), and we answered with a reference in the last revision. Further studies will certainly analyze both RNA and protein expression based on the reviewers' kind comments. We appreciated that your time and consideration.
(1) Wang, Z.; Li, B.; Jiang, H.; Ma, Y.; Bao, Y.; Zhu, X.; Xia, H.; Jin, Y. IL-8 exacerbates alcohol-induced fatty liver disease via the Akt/HIF-1α pathway in human IL-8-expressing mice. Cytokine 2021, 138, 155402.
※ We would like to thank the reviewers for the constructive and insightful comments and hope that our answers will be acceptable to the reviewers.
Reviewer 3 Report
The manuscript was re-reviewed for publication in the journal. The manuscript was designed to evaluate the effects of Bifidobacterium animalis ssp. lactis MG741 on nonalcoholic fatty liver disease (NAFLD) and weight gain using a mouse model of high-fat diet (HFD)-induced obesity. The results obtained indicate that the oral administration of MG741 could prevent NAFLD and obesity, thereby improving metabolic health. It is the reviewer’s opinion that the manuscript is easy to follow for the readers and the study is interesting. The authors promptly explained/discussed all issues suggested. However, it appears that there are still a couple of minor concerns in the manuscript. I believe that the manuscript is ready for the publication after the revision.
1) How about further low dose of 1.0 × 104? Because low dose may be better for clinical application.
2) The authors revised the steatosis score in Figure 3B. However, it appears that a dot graph may be better instead of a bar graph because a score is only 0, 1, 2, 3, and 4. Also, the authors should promptly consider a statistic analysis.
Author Response
Reviewer 3:
The manuscript was re-reviewed for publication in the journal. The manuscript was designed to evaluate the effects of Bifidobacterium animalis ssp. lactis MG741 on nonalcoholic fatty liver disease (NAFLD) and weight gain using a mouse model of high-fat diet (HFD)-induced obesity. The results obtained indicate that the oral administration of MG741 could prevent NAFLD and obesity, thereby improving metabolic health. It is the reviewer’s opinion that the manuscript is easy to follow for the readers and the study is interesting. The authors promptly explained/discussed all issues suggested. However, it appears that there are still a couple of minor concerns in the manuscript. I believe that the manuscript is ready for the publication after the revision.
1) How about further low dose of 1.0 × 104? Because low dose may be better for clinical application.
▶ A study of a low dose of 1.0 × 104 CFU has not yet been conducted. Even at 1.0 x 105 CFU, only a slight effect was shown, but not significant effects as much as 1.0 x 106 CFU. So, we were planning to do additional research only with higher dose than that.
2) The authors revised the steatosis score in Figure 3B. However, it appears that a dot graph may be better instead of a bar graph because a score is only 0, 1, 2, 3, and 4. Also, the authors should promptly consider a statistic analysis.
▶ Thank you for your kind comment. It was revised with a dot graph according to the reviewer's opinion. However, since the score is only 0, 1, 2 and 3, the dots seem to overlap. Statistical analysis showed the same result as last time as a result of re-analysis.
Line 206:
※ We would like to thank the reviewers for the constructive and insightful comments and hope that our answers will be acceptable to the reviewers.
Reviewer 4 Report
The authors answered to the questions properly and honestly that they didn't perform the experiments asked by the reviewer. As the title mentioned "gut-liver axis", the reviewer would expect to see more evidence to support this statement including changes of gut microbiota, changes of some gut-derived metabolites via the hepatoenteric circulation such as bile acids, etc. Since these experiments were not done, the reviewer would suggest to narrow the scope of the title to "Bifidobacterium animalis ssp. lactis MG741 reduces body weight and ameliorates nonalcoholic fatty liver disease via improving the gut permeability and amelioration of inflammatory cytokines."
Author Response
We highly appreciate the reviewer’s constructive and helpful comments on our manuscript. As suggested by the reviewer, we have carefully response (marked in blue) to address the reviewer’s comments and revised manuscript (marked in red). We hope that the reviewer will find our responses to the comments satisfactory.
Reviewer 4:
The authors answered to the questions properly and honestly that they didn't perform the experiments asked by the reviewer. As the title mentioned "gut-liver axis", the reviewer would expect to see more evidence to support this statement including changes of gut microbiota, changes of some gut-derived metabolites via the hepatoenteric circulation such as bile acids, etc. Since these experiments were not done, the reviewer would suggest to narrow the scope of the title to "Bifidobacterium animalis ssp. lactis MG741 reduces body weight and ameliorates nonalcoholic fatty liver disease via improving the gut permeability and amelioration of inflammatory cytokines."
▶ We appreciated your kind comment and revised title to your suggestion as below;
Lines 2–5: Bifidobacterium animalis ssp. lactis MG741 reduces body weight and ameliorates nonalcoholic fatty liver disease via improving the gut permeability and amelioration of inflammatory cytokines
※ We would like to thank the reviewers for the constructive and insightful comments and hope that our answers will be acceptable to the reviewers.
